# *SmRAV1*, an AP2 and B3 Transcription Factor, Positively Regulates Eggplant’s Response to Salt Stress

**DOI:** 10.3390/plants12244174

**Published:** 2023-12-15

**Authors:** Junjie Ding, Bowen Yao, Xu Yang, Lei Shen

**Affiliations:** College of Horticulture and Landscape Architecture, Yangzhou University, Yangzhou 225009, China; djj0211142023@163.com (J.D.); ybw2836988862@163.com (B.Y.); yangxu@yzu.edu.cn (X.Y.)

**Keywords:** eggplant (*Solanum melongena*), AP2 and B3 domain-containing transcription factor, *SmRAV1*, salt stress

## Abstract

Salt stress is a lethal abiotic stress threatening global food security on a consistent basis. In this study, we identified an AP2 and B3 domain-containing transcription factor (TF) named *SmRAV1,* and its expression levels were significantly up-regulated by NaCl, abscisic acid (ABA), and hydrogen peroxide (H_2_O_2_) treatment. High expression of *SmRAV1* was observed in the roots and sepal of mature plants. The transient expression assay in *Nicotiana benthamiana* leaves revealed that SmRAV1 was localized in the nucleus. Silencing of *SmRAV1* via virus-induced gene silencing (VIGS) decreased the tolerance of eggplant to salt stress. Significant down-regulation of salt stress marker genes, including *SmGSTU10* and *SmNCED1,* was observed. Additionally, increased H_2_O_2_ content and decreased catalase (CAT) enzyme activity were recorded in the *SmRAV1*-silenced plants compared to the TRV:*00* plants. Our findings elucidate the functions of *SmRAV1* and provide opportunities for generating salt-tolerant lines of eggplant.

## 1. Introduction

Salinity/salt stress is the second-biggest abiotic factor affecting agricultural productivity worldwide by damaging numerous physiological, biochemical, and molecular processes [1,2,3,4,5,6,7]. Breeding salt-tolerant varieties would certainly minimize the damaging effects of salinity [8]. To breed new verities, it is imminently important to under the intricate signaling pathway regulating plant response to salt stress. Despite the plethora of literature available, research gaps are still at large in the field of salt stress biology. Understanding the transcription factors mediating the underlying mechanism of salt stress would add another cornerstone to the scientific research.

Transcription factors play vital roles in plant growth and plant responses to abiotic stresses [9,10]. The RAV transcription factors contain a B3 DNA-binding domain and/or an APETALA2 (AP2) DNA-binding domain [11,12,13]. RAV transcription factors regulate the transcript expression of downstream target genes by binding to core sequences of 5′-CAACA-3′ and 5′-CACCTG-3′ [14] and function in regulating vegetative growth [15,16,17], flowering [11,18,19], phytohormone signal transduction [20,21], biotic stresses response [22,23,24,25], and abiotic stresses response such as salt [21,26,27,28], and drought [21,29]. To date, *RAV* gene family members have been identified in multiple plant species, such as six *RAV* genes in *Arabidopsis thaliana* [13], 26 in wheat [30], 33 in *Gossypium hirsutum* [31], 11 in pear [32], 15 in rice [33], 13 in soybean [34], and five in peach [35]. RAV proteins have been reported in *Arabidopsis thaliana,* garnering seed germination [17], seedling development [17], salt stress response [21,28], drought stress response [21], and flowering [36]. For example, Arabidopsis RAV1 interacts with SnRK2 kinases to repress the transcript expression of *ABI3*, *ABI4*, and *ABI5,* promoting seed germination and seedling development in an abscisic acid (ABA)-dependent manner [17]. The overexpression of Arabidopsis *RAVs* (*RAV1*, *RAV1L*, *RAV2*) decreased the tolerance of Arabidopsis plants to drought and salt stresses in an ABA-independent manner [21]. The *CmTEM1* overexpression delayed chrysanthemum flowering by directly binding to the flowering integrator *CmAFL1* promoter, thus repressing this gene’s expression [11]. In addition, RAV transcription factors also played an important role against pathogenic microorganism attacks in plants. The RNA-silencing suppressor (RSS) 24-kDa protein (p24) of grapevine leafroll-associated virus 2 repressed the direct activation of RAV transcription factor *VvRAV1* in *Vitis vinifera* to grapevine pathogenesis-related protein 1 (*PR1*) expression by interacting with VvRAV1, leading to a weakened resistance of grape to pathogens [37]. Although some reports indicated that RAV transcription factors played a vital role in plant against salt stress, the functions and mechanisms of RAV transcription factors against salt stress remain largely unclear.

Eggplant (*Solanum melongena*) is a popular *Solanaceae* vegetable that is sensitive to salinity stress [38]. Soil salinity significantly affects eggplant growth, development, and overall yield. The topics of salinity and seeking salt-tolerant cultivars are of great interest to agronomists and growers. Therefore, it is helpful to breed new eggplant cultivars with high salinity tolerance. In this study, we identified a RAV transcription factor family member named *SmRAV1*, which was significantly up-regulated by salt stress treatment, by analyzing mRNA sequencing data of eggplant roots treated with salt stress according to our previous study [39]. We further analyzed *SmRAV1* expression and the function in eggplant against salt stress, and demonstrated that *SmRAV1* positively regulates the tolerance of eggplant to salt stress.

## 2. Results

### 2.1. Sequence Analysis of SmRAV1

From our previous sequencing data, we observed that the *SmRAV1* (Smechr1102045) increased significantly in response to salt stress [39] (Appendix A). The *SmRAV1* has an ORF (open reading frame) of 1101 bp (base pair) and encoding a 366 bp amino acid. By utilizing the SMART (http://smart.embl-heidelberg.de/, accessed on 11 August 2023) website, we found that SmRAV1 possesses one AP2 and one B3 conserved domain (Figure 1a,d). We further analyzed the secondary and tertiary structures of SmRAV1 via the PRABI-GERLAND (https://npsa-prabi.ibcp.fr/cgi-bin/npsa_automat.pl?page=/NPSA/npsa_sopma.html, accessed on 11 August 2023) and SWISS-MODEL (https://swissmodel.expasy.org/interactive, accessed on 11 August 2023) websites, respectively. The results showed that the secondary structure of SmRAV1 consisted of four types, including alpha helix (Hh), beta turn (Tt), extended strand (Ee), and random coil (Cc) (Figure 1b, Appendix A). The results of tertiary structures of SmRAV1 showed that it shared 82.45% similarities with the E1U2K4.1.A tertiary structure model (Figure 1c). We next performed the multiple alignment of amino acid sequences and found that the SmRAV1 amino acid sequence shared 82, 78, 78, 78%, 76, 69, 64, 61, 61, 53, 58, and 52% sequence similarities with its homologs from the other plant species, including *Solanum dulcamara* SdRAV1-like (XP_055801403.1), *Solanum tuberosum* StRAV1 (XP_006366067.1), *Solanum stenotomum* StRAV1-like (XP_049391936.1), *Solanum verrucosum* SvRAV1-like (XP_049363435.1), *Solanum lycopersicum* SlRAV1 (XP_004236999.1), *Nicotiana tabacum* NtRAV1-like (XP_016445426.1), *Vitis vinifera* VvRAV1 (UYF10651.1), *Solanum lycopersicum* SlRAV2 (ABY57635.1), *Capsicum annuum* CaRAV1 (XP_016548054.2), *Zea mays* ZmRAV1 (PWZ33711.1), *Arabidopsis thaliana* AtRAV1 (OAP16099.1), and *Oryza sativa* Os01g0693400 (NP_001388398.1), respectively (Figure 1d). Moreover, we constructed evolutionary trees by using their amino sequences to analyze the phylogenetic relationships between the above RAV homologs. SmRAV1 had the closest relationship with SdRAV1-like (Figure 1e). In addition, we predicted the *cis*-elements within the 2000 bp promoter of *SmRAV1* via the PlantCARE (http://bioinformatics.psb.ugent.be/webtools/plantcare/html/, accessed on 12 August 2023) website. The *SmRAV1* contained multiple *cis*-elements related to phytohormone response (two ethylene response elements (EREs) and one gibberellin response element (GARE-motif), transcription factor-binding elements (five MYB transcription factor-binding elements (MYB), four MYC transcription factor-binding elements (MYC), two WRKY transcription factor-binding elements (W-box), one bZIP transcription factor-binding element (A-box), and one HD-zip transcription factor binding element (HD-Zip), and stress response element STRE, apart from the elements related to light response (Figure 1f).

### 2.2. Expression Profiles of SmRAV1 under Different Stresses

The expression of *SmRAV1* in the roots of salt-stressed eggplant was analyzed. We found that the expression level of *SmRAV1* was significantly up-regulated by salt stress treatment, and reached its peak at 24 h. We also investigated the expression profiles of *SmRAV1* under different treatments, including high temperature (HT), low temperature (LT), dehydration stress, ABA, and hydrogen peroxide (H_2_O_2_), in eggplant. The result showed that apart from the NaCl treatment, both the ABA and H_2_O_2_ treatments could significantly induce the up-regulation of *SmRAV1* expression. On the other hand, low expression was recorded in response to HT, LT, and dehydration stress, respectively (Figure 2), implying that *SmRAV1* may function in eggplant response to salt stress or be involved in ABA- and H_2_O_2_-mediated signal transduction pathways.

### 2.3. Analysis of Tissue Specific Expression of SmRAV1

The tissue-specific expression was observed in order to understand the role of *SmRAV1* in developmental biology. A significantly higher expression was observed in the roots compared to young leaves (YL) and stems (ST). Also, high expression was recorded in the sepals (SE) and flowers (FL) than other tissues (Figure 3).

### 2.4. Subcellular Localization of SmRAV1

The conserved domain analysis revealed that *SmRAV1* was composed of AP2 and B3 domains (Figure 1a). SmRAV1 proteins may localize in the nucleus. For validation, we firstly predicted SmRAV1 subcellular localization by searching the Plant-mPLoc (http://www.csbio.sjtu.edu.cn/bioinf/plant-multi/, accessed on 12 August 2023) website using an SmRAV1 amino acid sequence [40]. The result showed that SmRAV1 proteins localized in the nucleus (Appendix A). We searched the INSP (http://www.csbio.sjtu.edu.cn/bioinf/INSP/, accessed on 12 August 2023) website using the amino acid sequence, which further confirmed the nuclear signals of SmRAV1 [41]. SmRAV1 has an NLS (_201_VGKLNRLV_208_) in its amino acid sequence (Figure 1d, Appendix A). These results suggest that SmRAV1 is probably a nuclear-localized protein. Subsequently, we performed a subcellular localization assay. We constructed the recombinant vector 35S:*GFP-SmRAV1* by cloning the *SmRAV1* coding sequence (CDS) into plant overexpression vector pBinGFP2. The CDS of *AtH2B* was fused with a red fluorescence protein (RFP) tag to generate 35S:*AtH2B-RFP* structure. The AtH2B is generally used as a nuclear localization marker [42,43] (Figure 4a). The *Agrobacterium* GV3101 cells harboring the 35S:*AtH2B-RFP* structure were mixed with the GV3101 cells containing the 35S:*GFP-SmRAV1* or 35S:*GFP* (empty vector) structures at a 1:1 ratio, respectively. The *Nicotiana benthamiana* leaves infiltrated with above *Agrobacterium* mixtures were used to observe the fluorescence signals using a laser scanning confocal microscope. We found that the green fluorescence signals of SmRAV1-GFP proteins occurred in the nucleus of the epidermic cells in *Nicotiana benthamiana* leaves, and overlapped with the red nuclear fluorescence signals of localization marker AtH2B-RFP proteins. Meanwhile, the green fluorescence signals of GFP proteins expressed by 35S:*GFP* vector were distributed in entire cells (Figure 4b), suggesting that SmRAV1 localizes in the nucleus.

### 2.5. Silencing of SmRAV1 Enhances Susceptibility of Eggplant against Salt Stress

Previous studies have revealed that RAV transcription factors play vital roles against salt stress in plants [21,27,30,44]. In this study, we also found that salt stress could significantly induce *SmRAV1* expression in eggplant roots (Figure 2). Therefore, we speculated that *SmRAV1* may play an essential role in eggplant’s tolerance against salt stress. To confirm this possibility, we performed a virus-induced gene silencing (VIGS) assay to silence *SmRAV1* expression. Firstly, we detected the silencing efficiency of *SmRAV1* by RT-qPCR assay. The transcription level of *SmRAV1* in the roots of *SmRAV1*-silenced (TRV:*SmRAV1*) eggplant plants was significantly reduced (approximately 63%) compared to the control plants (TRV:*00*) under salt stress treatment (Figure 5a). At 36 h post-treatment, the *SmRAV1*-silenced eggplant exhibited more serious wilting symptoms than the control plants under 200 mM NaCl solution treatment (Figure 5b), accompanied by a lower survival rate (Figure 5c). In addition, reduced expression of *SmGSTU10* and *SmNCED1* (Figure 5d), markedly decreased reactive-oxygen-scavenging enzymes catalase (CAT) activity, and increased H_2_O_2_ content under salt stress treatment were also observed (Figure 5e). These results suggest that *SmRAV1* positively modulates the response of eggplant to salinity stress.

## 3. Discussion

Transcription factors are proteins that help to turn specific genes “on” or “off” by binding to nearby DNA [45,46,47,48,49]. Accumulating evidence has revealed the various functions of RAV transcription factors in the processes of seed germination [17,21,29], seedling growth and development [16,17,18,50], flowering [11,19], and abiotic and biotic stresses [22,23,24,29,34,51]—salt stress in particular [26,27,28,30,50]. Among the multiple abiotic stresses, salt stress is one of the most important factors threating plants during their entire growth processes. Therefore, it is important to improve the salt tolerance through molecular and biochemical approaches. Although previous studies have reported the functions and regulatory mechanisms of RAV transcription factors, the underlying functions and regulatory mechanisms of RAV transcription factors in plants’ responses to salt stress remain elusive. Herein, we identified a RAV transcription factor family member named *SmRAV1* and examined its expression profiles under HT, LT, salt, dehydration, ABA, and H_2_O_2_ treatment; subcellular localization; and the function of *SmRAV1* in eggplant’s response to salt stress.

By analyzing the mRNA sequencing data of eggplant roots treated with salt stress [39], we obtained a candidate gene named *SmRAV1*, which had the highest sequence similarity with its homolog *RAV1* in *Arabidopsis thaliana*, and its expression was up-regulated by salt stress (Appendix A). To confirm this result, an induced expression profile of *SmRAV1* in eggplant roots treated with NaCl solution was observed (Figure 2). Arabidopsis RAV1 negatively modulates the tolerance of Arabidopsis to drought and salt stress [21]. We analyzed *SmRAV1* sequences and found that SmRAV1 harbored one conserved AP2 and B3 domain (Figure 1a), and exhibited the highest sequence similarity with SdRAV1-like in *Solanum dulcamara* (Figure 1d,e). However, no reports exist of *SdRAV1*-like involvement in *Solanum dulcamara’s* response to salt stress. In addition, we found some *cis*-elements related to phytohormone response and transcription factor binding within the *SmRAV1* promoter. These *cis*-elements were related to the physiological and biochemical processes, as well as the stress signal response [52,53,54]. Therefore, we presumed that *SmRAV1* may function in eggplant’s response to salt stress. *SmRAV1* expression was not only induced by salt stress, but also up-regulated by ABA and H_2_O_2_ treatment, although it was significantly down-regulated by HT, LT, and dehydration treatment (Figure 2), indicating that *SmRAV1* may play a negative role in eggplant’s response to HT, LT, and dehydration stresses. These results are similar to those of the previous reports. For example, *NtRAV4* played a negative role in *Nicotiana tabacum* against drought stress by enhancing its antioxidant capacity and defense system [55]. In Arabidopsis, *RAV1* expression was reduced by dryness, and its overexpression enhanced transpiration in drought conditions, thus enhancing sensitivity [21]. Moreover, we detected the expression levels of *SmRAV1* in diverse tissues from eggplant plants in different developmental stages. High expression was found in the roots of seedlings as well as in the SE and FL of mature plants (Figure 3), suggesting that *SmRAV1* may be involved in the process of flower development or the growth and defense of roots. SmRAV1 proteins may localize in the nucleus by predicting its subcellular localization and NLS sequences (Appendix A). We further verified that SmRAV1 proteins localize in the nuclei of epidermic cells of *Nicotiana benthamiana* leaves by subcellular localization assay (Figure 4b).

To investigate the function of *SmRAV1* in eggplant’s response to salt stress, we carried out a VIGS assay to assess the effect of *SmRAV1* silencing and the tolerance of eggplant against salt stress. Our data demonstrated that the silencing of *SmRAV1* decreased the tolerance of eggplant to salt stress (Figure 5b) and markedly down-regulated the expression level of salt stress defense related marker genes (*SmGSTU10* and *SmNCED1*) (Figure 5d), which was accompanied by a significant decrease in CAT enzyme activity and an increase in H_2_O_2_ content (Figure 5e). It can be suggested that *SmRAV1* positively functions in eggplant against salt stress. Similar to our findings, overexpression of *MtRAV3* was found to enhance the tolerance of *Medicago truncatula* against salt and osmotic stresses [50]. Wheat *RAV1* overexpression enhanced the tolerance of Arabidopsis to salt stress [30]. In *Betula platyphylla*, BpRAV1 could activate the expression of the *superoxide dismutase* (*SOD*) and *peroxidase* (*POD*) genes by directly binding to the RAV1A and RBS1 elements to enhance the reactive-oxygen-species-scavenging abilities of SOD and POD, thus improving salt and osmotic stress tolerance [26]. A recent study showed that *CsRAV1* overexpression enhanced the tolerance of transgenic Arabidopsis and cucumber seedlings to salt stress and ABA [44]. Interestingly, few reports have indicated that RAV transcription factors acted as negative regulators of salt stress. For example, Arabidopsis *RAV1*, *RAV1L* and *RAV2*/*TEM2* negatively regulated its tolerance to salt and drought stress [21]. The overexpression of *A-TsRAVs*, RAV proteins divided into group A according to their sequence similarity in *Thellungiella salsuginea*, showed weak growth retardation under salt stress conditions, suggesting that all *A-TsRAVs* play a negative role in Arabidopsis against salt stress [56]. It is worth mentioning that, apart from the function of RAV transcription factors in the fine-tuning of plant responses to abiotic stress, RAV proteins seem to be negative regulators in plant growth and development [18,21,56,57].

In a word, our data indicate that *SmRAV1* positively functions in eggplant’s response to salt stress and lays a foundation for the genetic enhancement of the salinity tolerance of eggplant.

## 4. Materials and Methods

### 4.1. Plant Material Growth and Conditions

The growth conditions of eggplant and *Nicotiana benthamiana* plants followed our previous studies [39,58]. Briefly, seeds of the inbred eggplant line ML41 were packaged with clean gauze, then soaked in a 55 °C water bath for 15 min and left in tap water at room temperature overnight. The seeds were placed in an illumination incubator under the condition of 25 °C for germination. The seedlings with two cotyledons were planted in small plastic pots with nutrient soil, then placed into an illumination incubator under the conditions of 25 °C, a 16 h light/8 h dark photoperiod, and 60% relative humidity. Seeds of *Nicotiana benthamiana* were sowed on wet filter paper and placed in the illumination incubator at 25 °C for germination. The seedlings of *Nicotiana benthamiana* were planted in small plastic pots with nutrient soil, then grown under the same conditions as eggplant.

### 4.2. Abiotic Stresses Treatment

For the treatment of HT and LT, the 4–6-leaf-stage eggplant plants were placed in the illumination incubator under the conditions of 43 °C and 4 °C, respectively. The treated leaves were harvested under the liquid nitrogen condition at the time points of 0, 0.5, 1, 3, 6, and 12 h. The eggplant plants were gently pulled from the nutrient soil, and then the roots were washed with tap water. The washed plants were cultivated in Hoagland nutrient solution for two days. For NaCl, ABA, and H_2_O_2_ treatment, the roots were soaked in 200 mM NaCl, 100 μM ABA, and 1 mM H_2_O_2_ solution, respectively. The roots were harvested under the condition of liquid nitrogen at the time points of 0, 2, 6, 12, 24, and 48 h. For dehydration stress treatment, the surfaces of the eggplant roots were wiped and placed on the experiment table for continuous dehydration. The roots were harvested under liquid nitrogen at the time points of 0, 0.5, 1, 3, 6, and 9 h.

### 4.3. Plant Total RNA Extraction, cDNA Synthesis, and RT-qPCR Analysis

The samples were ground into powder under liquid nitrogen conditions, and the total RNA of the eggplant leaves or roots was extracted using the FastPure Plant Total RNA Isolation Kit (polysaccharides- and polyphenolics-rich) kit (RC401-01, Vazyme, Nanjing, China). The purity and concentration of total RNA were measured by an ultra-microspectrophotometer, and the integrity of the total RNA was assessed by an agarose gel electrophoresis assay. For cDNA synthesis, the mRNA from the total RNA was synthesized to cDNA using a HiScript III RT SuperMix for qPCR (+gDNA wiper) kit (R323-01, Vazyme). The transcript expression levels of the target genes were detected by RT-qPCR assay. The RT-qPCR assay was carried out using the ChamQ Universal SYBR qPCR Master Mix kit (Q711-02, Vazyme), according to the specifications. The specific primer pair sequences used to detect the target genes’ expression are listed in Appendix A. *SmActin* (Smechr1100649) was used as a reference gene to standardize the target genes’ expression. Three biological replications and the 2^−∆∆CT^ method were used to analyze the expression of the target genes [59].

### 4.4. Agrobacterium tumefaciens Cultivation and Infiltration

*Agrobacterium tumefaciens* strain GV3101 cells, harboring 35S: *GFP-SmRAV1* or 35S: *GFP* structures, were cultivated in liquid Luria–Bertani medium containing 50 μg/mL kanamycin and 50 μg/mL rifampicin in a shaker at 200 rpm and 28 °C overnight. The GV3101 cells were harvested by centrifuging at 28 °C and 6000 rpm for 5 min. The cells were resuspended in infiltration buffer (10 mM MES, 10 mM MgCl_2_, 200 mM acetosyringone, pH = 5.4) and the OD_600_ was adjusted to 0.8. The bacterial solution was incubated in the shaker at 28 °C and 60 rpm for 1~3 h, and then injected into the leaves using a disposable sterile syringe without a needle.

### 4.5. Subcellular Localization Assay

The full-length CDS of *SmRAV1* was cloned into the restriction endonuclease *Sma* Ⅰ of plant overexpression vector pBinGFP2 using the ClonExpress II One Step Cloning Kit (C112-01, Vazyme). *Agrobacterium tumefaciens* strain GV3101 cells harboring 35S:*AtH2B-RFP* vector were mixed with the GV3101 cells carrying 35S:*GFP-SmRAV1* or 35S:*GFP* structures at a 1:1 ratio, and then gently incubated in the shaker at 28 °C and 60 rpm for 1 h. The mixtures were infiltrated into the leaves of *Nicotiana benthamiana*. After 48 h, fluorescence signals were observed in the epidermic cells of *Nicotiana benthamiana* leaves via laser scanning confocal microscope with an excitation wavelength of 488 nm for GFP and an excitation wavelength of 532 nm for RFP.

### 4.6. VIGS Assay

The VIGS assay was performed following our previous studies [39,58]. The 300 bp specific DNA fragment of *SmRAV1* was amplified using specific primer pairs and cloned into the entry vector pDONR207 by a BP reaction, then transferred into the destination vector pTRV2:*00* by an LR reaction. The *Agrobacterium* GV3101 cells containing TRV1 vector were mixed with the cells harboring TRV2:*00*, TRV2:*SmRAV1*, or TRV2:*SmPDS* vectors at a 1:1 ratio, and then slowly incubated under the conditions of 28 °C and 60 rpm for 1 h. The mixtures were injected into the cotyledons of 2–3-week-old eggplant seedlings, and the infiltrated seedlings were placed into the illumination incubator at 20 °C without light for 48 h. After treatment, seedlings grew under the conditions of 25 °C, a 16 h light/8 h dark photoperiod, and a relative humidity of 60 for 3 weeks.

### 4.7. Measurement of Physiological Indices

The physiological indices, including CAT enzyme activity and H_2_O_2_ content in the roots of *SmRAV1*-silenced and control eggplant plants, were performed following the previous studies [39,58].

## 5. Conclusions

In this study, we selected a RAV transcription factor family member called *SmRAV1* by analyzing the mRNA sequencing data of eggplant roots treated with salt stress, and it was up-regulated by the salt stress. The results of further analysis showed that the transcript expression levels of *SmRAV1* were significantly up-regulated by salt, ABA, and H_2_O_2_ treatment, but down-regulated under HT, LT, and dehydration stress treatment. By means of a subcellular localization assay, the SmRAV1 protein localized in the nucleus. According to the results of the VIGS assay, we found that silencing of *SmRAV1* decreased the tolerance of eggplant to salt stress; significantly down-regulated the transcript expression levels of salt stress defense-related marker genes, including *SmGSTU10* and *SmNCED1*; markedly decreased CAT enzyme-activity; and increased H_2_O_2_ content compared to control plants. These results indicate that *SmRAV1* positively functions in eggplant’s response to salt stress, provide new insight into the function of RAV transcription factors in plants against salt stress.

## Figures and Tables

**Figure 1 plants-12-04174-f001:**
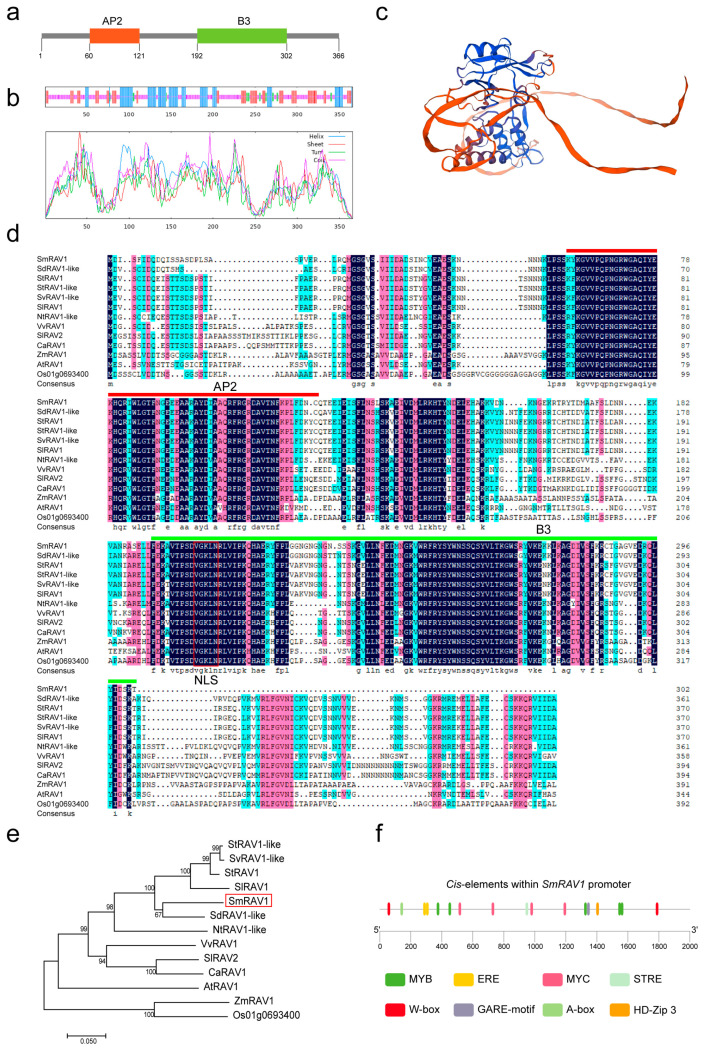
Sequence analysis of SmRAV1. (**a**) Schematic diagram of conserved domains of SmRAV1. (**b**) Analysis of secondary structure of SmRAV1. The blue, red, green, and azalein lines, respectively, represent secondary structure types of helix, sheet, turn, and coil. (**c**) Prediction of tertiary structure model of SmRAV1 protein. (**d**) Analysis of multiple alignment analysis of RAV proteins’ amino acid sequences from *Solanum melongena*, *Solanum dulcamara*, *Solanum tuberosum*, *Solanum stenotomum*, *Solanum verrucosum*, *Solanum lycopersicum*, *Nicotiana tabacum*, *Vitis vinifera*, *Capsicum annuum*, *Zea mays*, *Arabidopsis thaliana*, and *Oryza sativa*. The black, azalein, and blue shadows, respectively, represent 100%, 75~100%, and 50~75% sequence similarity. (**e**) Phylogenetic relationship analysis of SmRAV1 with its homologs from selected plant species. (**f**) Prediction of *cis*-elements within the assumed promoter sequences of *SmRAV1*.

**Figure 2 plants-12-04174-f002:**
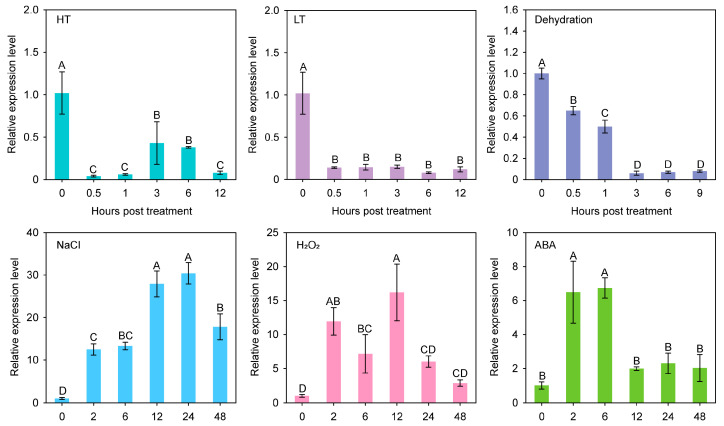
Analysis of transcript expression profiles of *SmRAV1* in eggplant leaves or roots under HT, LT, dehydration, NaCl, H_2_O_2_, and ABA treatment. Three biological repeats were used to calculate the mean ± standard deviation. Different upper letters indicate significant differences, as analyzed by Fisher’s protected LSD test (*p* < 0.01).

**Figure 3 plants-12-04174-f003:**
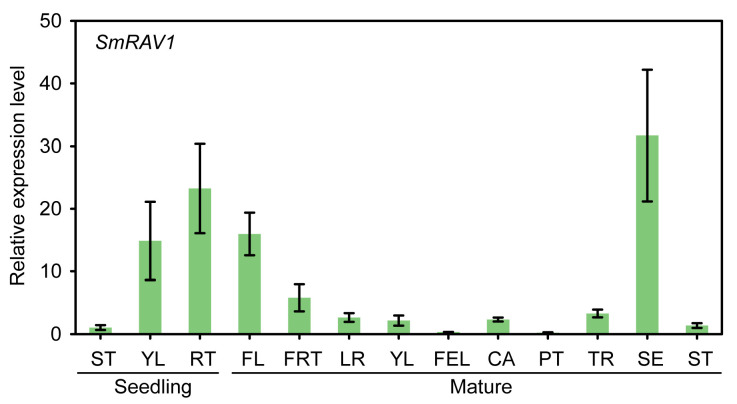
Analysis of tissue-specific expression of *SmRAV1*. Three biological repeats were used to calculate the mean ± standard deviation.

**Figure 4 plants-12-04174-f004:**
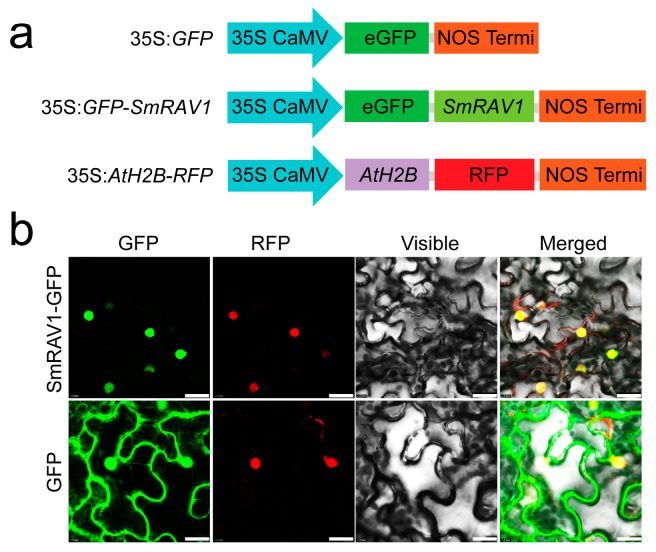
Subcellular localization of SmRAV1. (**a**) Schematic diagram of 35S:*GFP*, 35S:*GFP-SmRAV1*, and 35S:*AtH2B-RFP* structures. (**b**) Analysis of subcellular localization of SmRAV1 in the epidermic cells of *Nicotiana benthamiana* leaves. Bar = 25 μm.

**Figure 5 plants-12-04174-f005:**
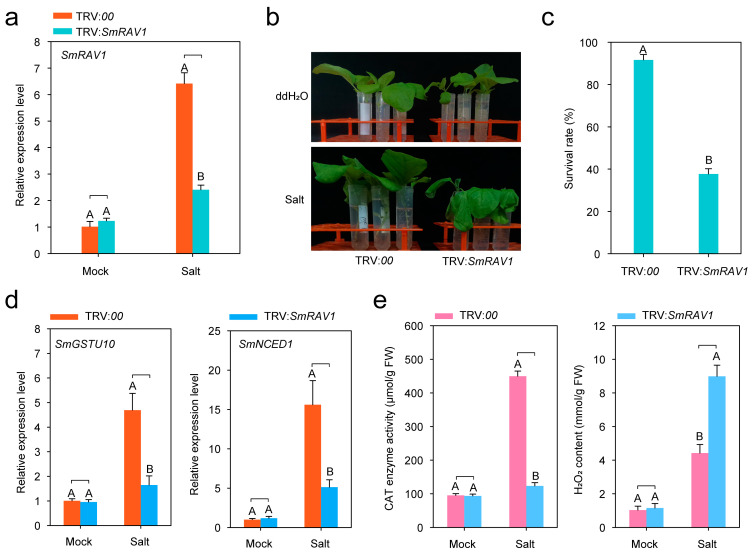
Silencing of *SmRAV1* enhances susceptibility of eggplant’s response to salt stress. (**a**) Detection of silencing efficiency of *SmRAV1*. (**b**) *SmRAV1*-silenced eggplant plants exhibited more sensitivity under salt stress treatment at 36 h post-treatment. (**c**) Analysis of survival rate of *SmRAV1*-silenced and control plants under salt stress treatment at 36 h post treatment. (**d**) Analysis of transcript expression levels of salt-stress-defense-related marker genes, including *SmGSTU10* and *SmNCED1,* in the roots of *SmRAV1*-silenced and control eggplants at 24 h post-salt-stress treatment. (**e**) Measurement of CAT enzyme activity and H_2_O_2_ content in the roots of *SmRAV1*-silenced and control eggplants at 48 h post-salt-stress treatment. In (**a**,**c**–**e**), three biological repeats were used to calculate the mean ± standard deviation. Different upper letters indicate significant differences, as determined by Student’s *t*-test (*p* < 0.01).

## Data Availability

Data are contained within the article and Appendix A.

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
