# Peer review of "SmRAV1, an AP2 and B3 Transcription Factor, Positively Regulates Eggplant’s Response to Salt Stress"

_plants, 2023, doi:10.3390/plants12244174_

Round 1
Reviewer 1 Report
Comments and Suggestions for Authors
A very interesting but complex piece of research that would be of interest mainly to those working at the molecular and transcription levels. Some of figures, although novel, would hard to interpret by non specialists
Of course the topic of salinity and seeking salt tolerant cvs would be of great interest to agronomists and growers. And this should be stressed
In the conclusion, the investigators should reflect as to other effects when salt tolerant cvs are developed.
Nevertheless the concusions should be simply presented.
How about future implications!
Is the quality of the fruit likely to change - texture, flavour, colour
Do the genetic changes protect against or worsen the effects of other abiotic stresses - temperature, drought, nutrition?
Do the genetic changes protect against or worsen the effects of other biotic stresses -infection and infestation?
Comments on the Quality of English Language
Satisfactory - but subject to spelling and grammar check
Author Response
Reviewer 1
A very interesting but complex piece of research that would be of interest mainly to those working at the molecular and transcription levels. Some of figures, although novel, would hard to interpret by non-specialists
Of course, the topic of salinity and seeking salt tolerant cvs would be of great interest to agronomists and growers. And this should be stressed
Response: Thank you very much for your good suggestion. We added this content, please see in lines 72-73 in the revised manuscript.
In the conclusion, the investigators should reflect as to other effects when salt tolerant cvs are developed.
Nevertheless the conclusions should be simply presented.
Response: Thank you very much for your good suggestion. We presented the conclusions in the section 5, please see in lines 337-351 in the revised manuscript.
How about future implications!
Response: Thank you very much. Our data will lay a foundation to breed new eggplant cultivars with high salinity-tolerance by exploring to the function of SmRAV1 in eggplant against salt stress.
Is the quality of the fruit likely to change - texture, flavour, colour
Response: Thank you very much. Our data could not directly demonstrate whether silencing or overexpression of SmRAV1 in eggplant affects the texture, flavour, and colour of eggplant fruits.
Do the genetic changes protect against or worsen the effects of other abiotic stresses - temperature, drought, nutrition?
Response: Thank you very much. Our data could not directly demonstrate whether genetic changes (silencing of SmRAV1) in eggplant protect against or worsen the effects of other abiotic stresses - temperature, drought, nutrition.
Do the genetic changes protect against or worsen the effects of other biotic stresses -infection and infestation?
Response: Thank you very much. Our data could not directly demonstrate whether genetic changes (silencing of SmRAV1) in eggplant protect against or worsen the effects of other biotic stresses -infection and infestation.
Reviewer 2 Report
Comments and Suggestions for Authors
Please see the attached comments.

Author Response
Reviewer 2
AP2 and B3 Domain-Containing Transcription Factor SmRAV1 Positively Functions in Eggplant Response to Salt Stress
In this study, the researchers identified and focused on a specific member of the RAV transcription factor family, denoted as SmRAV1. The selection was based on the analysis of mRNA sequencing data from eggplant roots subjected to salt stress, revealing an up-regulation of SmRAV1 under these conditions. Further investigation demonstrated that the expression levels of SmRAV1 were notably increased in response to salt, abscisic acid (ABA), and hydrogen peroxide (H2O2) treatments. Conversely, the expression was down-regulated when the plants were exposed to high temperature (HT), low temperature (LT), and dehydration stress.
- The manuscript is well written, however, the expression of the transcription factor under different stressors is not solid enough. Please provide how the applied concentrations and sampling times were selected?
Response: Thank you very much. The applied concentrations of 200 mM NaCl, 100 μM ABA, and 1 mM H2O2 solution and sampling times were selected according to the previous studies described [1, 2].
The upregulation of SmRAV1 expression is associated with salt stress and coincides with the overaccumulation of H2O2, indicating a connection to reactive oxygen species. In contrast, dehydration induces osmotic stress, leading to a water deficit and changes in cellular water potential. The lack of significant SmRAV1 induction under dehydration stress implies that SmRAV1 may not be primarily responsive to osmotic stress alone. Notably, salt stress encompasses not only osmotic stress but also ion toxicity and the accumulation of reactive oxygen species, including H2O2. The observed increase in SmRAV1 expression under salt stress, coupled with H2O2 overaccumulation, suggests a potential linkage between SmRAV1 and the oxidative stress component associated with salt stress.
- Given the association of SmRAV1 with H2O2 accumulation, what is the specific role of SmRAV1 in regulating reactive oxygen species (ROS) during salt stress?
Response: Thank you very much. According to our data, we found that silencing of SmRAV1 decreased the tolerance of eggplant against salt stress, accompanied by the significant increase of H2O2 content and decrease of CAT enzyme activity, implying that SmRAV1 may activate the expression of down-stream salt stress defense related genes to enhance the enzyme activities of reactive-oxygen-scavenging enzymes such as CAT to get rid of excessive ROS in the process of plant response to salt stress. We need further evidences to explain how SmRAV1 regulates the level of ROS in eggplant response to salt stress.
- Please explain whether there are specific genes or pathways identified in this study that respond specifically to ion toxicity induced by salt stress?
Response: Thank you very much. In this study, we have not identified the specific genes or pathways that respond specifically to ion toxicity induced by salt stress. According to our data, we found that Silencing of SmRAV1 significantly down-regulated the expression of the salt stress defense related gene SmGSTU10, which encodes a Tau class glutathione S-transferase protein. Previous studies suggest that glutathione S-transferase proteins play important roles in plant response to abiotic stresses via detoxification and antioxidant response to reduce the damage of adverse environmental factors or oxidative free radicals to plant cells [3-5]. Therefore, SmGSTU10 may be a specific gene to involve in the specific response to ion toxicity induced by salt stress in eggplant.
References
[1] L. Shen, X. Xia, L. Zhang, S. Yang, X. Yang, SmWRKY11 acts as a positive regulator in eggplant response to salt stress, Plant Physiology and Biochemistry 205 (2023) 108209.
[2] S. Li, N. Wang, D. Ji, W. Zhang, Y. Wang, Y. Yu, S. Zhao, M. Lyu, J. You, Y. Zhang, L. Wang, X. Wang, Z. Liu, J. Tong, L. Xiao, M.Y. Bai, F. Xiang, A GmSIN1/GmNCED3s/GmRbohBs Feed-Forward Loop Acts as a Signal Amplifier That Regulates Root Growth in Soybean Exposed to Salt Stress, Plant Cell 31(9) (2019) 2107-2130.
[3] X. Li, Y. Pang, Y. Zhong, Z. Cai, Q. Ma, K. Wen, H. Nian, GmGSTU23 Encoding a Tau Class Glutathione S-Transferase Protein Enhances the Salt Tolerance of Soybean (Glycine max L.), Int J Mol Sci 24(6) (2023).
[4] B. Jha, A. Sharma, A. Mishra, Expression of SbGSTU (tau class glutathione S-transferase) gene isolated from Salicornia brachiata in tobacco for salt tolerance, Mol Biol Rep 38(7) (2011) 4823-32.
[5] E. Horvath, K. Bela, A. Galle, R. Riyazuddin, G. Csomor, D. Csenki, J. Csiszar, Compensation of Mutation in Arabidopsis glutathione transferase (AtGSTU) Genes under Control or Salt Stress Conditions, Int J Mol Sci 21(7) (2020).